# Protocol for evaluating the cost-effectiveness of Mongolia's sugar-sweetened beverages tax using double machine learning

**Nyamdavaa Byambadorj**[1]*, **Rohan Best**[1], **Undram Mandakh**[2], **Kompal Sinha**[1]

1 Department of Economics, Macquarie Business School, Macquarie University, Sydney, New South Wales, Australia, 2 Department of Family Medicine, School of Medicine, Mongolian National University of Medical Science, Ulaanbaatar, Mongolia

* dibo.byambadorj@mq.edu.au

**Data availability statement:** This study does not currently include pilot or preliminary data.

## Abstract

Elevated consumption of sugar-sweetened beverages (SSBs) has been associated with an increase in obesity, type 2 diabetes, and other non-communicable diseases (NCDs), a significant health and economic burden on Mongolia. To address this, the government has introduced a 20% SSB tax set to take effect in 2027. This study conducts a Cost-Effectiveness Analysis (CEA) using a Markov cohort model, incorporating Double Machine Learning (DML) to estimate price elasticity and assess policy-driven consumption changes while addressing potential confounding. The analysis integrates DML-estimated price elasticity and consumption shifts with disease transition probabilities, simulating outcomes for the 2023 Mongolian population, aged over 15 years old, over two time horizons of 20 years and a lifetime. The model estimates changes in obesity prevalence, healthcare costs, and disease burden, translating them into Disability-Adjusted Life Years (DALYs) averted, and Quality-Adjusted Life Years (QALYs) gained. Tax revenue projections and sensitivity analyses further assess the robustness of assumptions. By combining machine learning-based causal inference with economic modelling, this study provides policy-relevant evidence on the cost-effectiveness of SSB taxation, supporting data-driven decision-making for public health strategies in Mongolia, highlighting the tax's potential to reduce the burden of NCDs and promote healthier behaviours.

## Introduction

The global increase in the consumption of sugar-sweetened beverages (SSBs) presents a serious public health issue, strongly related to chronic diseases such as obesity, type 2 diabetes, cardiovascular disease (CVD), and dental caries [1–3]. High in calories and low in essential nutrients, SSBs contribute to weight gain, metabolic disorders, and related complications, significantly when not offset by physical activity [1]. Higher consumption of SSBs is associated with a higher incidence of type 2 diabetes, with each serving per day increasing the risk

All analyses outlined in this protocol will rely on secondary datasets, including: National Nutrition Survey (2023) managed by the National Center for Public Health. Access can be requested through the National Center for Public Health. STEP Survey (2019): Managed by the Public Health Institute. Access can be requested through https://extranet.who.int/ncdsmicrodata/index.php/catalog/836. Household Socioeconomic Survey: Managed by the National Statistical Office. Access can be requested through http://web.nso.mn/nada/index.php/catalog/HSES/dataset. Mongolian Burden of Disease Study: Offers comprehensive data on disease epidemiology and mortality rates, including background mortality. Access can be requested through https://www.healthdata.org/research-analysis/health-by-location/profiles/mongolia Population Data: Sourced from the National Statistical Office, access can be requested through www.1212.mn. The authors do not have permission to share these datasets directly. Interested researchers may obtain access by applying to the respective organisations.

**Funding:** The author(s) received no specific funding for this work.

**Competing interests:** The authors have declared that no competing interests exist.

by 18% [4]. SSBs contribute to dental caries in children, straining dental health systems [5]. These health burdens are especially significant in middle-income countries, where obesity rates increase in parallel with increased SSB consumption [6]. Recognising these challenges, policymakers and the World Health Organization advocate for price-based interventions, such as taxation, to curb SSB consumption and its associated health risks.

Global studies consistently show that SSB taxes reduce consumption, improve public health, and generate revenue for health initiatives [7–10]. A meta-analysis of 62 global studies revealed that SSB demand is highly responsive to tax-induced price increases, with a price elasticity of -1.59 and an average reduction in SSB sales of 15% [11]. Decreased purchases caused by high price elasticity have also been confirmed in several studies in developed countries including the United States of America [12], and Canada [13]. Systematic reviews across multiple countries indicate that SSB taxes can reduce the incidence and mortality of non-communicable diseases (NCD) while generating substantial government revenue and healthcare savings that exceed implementation costs [14–16]. These findings suggest that SSB taxes are crucial in combating worldwide obesity and NCDs. However, there remains a critical research gap in low- and middle-income countries, underlining the need for focused research in these settings.

Like many low- to middle-income countries, Mongolia faces unique challenges as rising urbanisation, changing lifestyles, and aggressive marketing drive SSB consumption. Limited access to fresh foods further exacerbates NCD burdens, particularly among children and adolescents [17]. Forty-five percent of the population consumes excessive carbohydrates, with 20% of adults classified as obese and 26% as overweight [18]. In 2017, Mongolia's obesity-related death rate was three times the global average, with cardiovascular diseases contributing to 55% of hospitalisations and 37% of deaths [19]. Mongolia ranks among the top 20 countries for childhood obesity, with 81% of school children consuming SSBs weekly [19]. Policymakers have approved a 20% tax on SSBs to address rising concerns, scheduled for 2027. However, there has been limited examination of the potential health and economic outcomes of the policy. This research aims to fill this gap by assessing the cost-effectiveness of the proposed SSB tax in Mongolia by applying a novel analytical approach. A Markov cohort model, combined with Double Machine Learning (DML), will enhance causal inference by controlling for confounding factors and capturing complex, non-linear relationships. DML will improve the reliability of estimates for health outcomes, healthcare savings, and tax revenue [20]. Key outcomes include estimates of health benefits, healthcare cost savings, and tax revenue. The analysis will evaluate socioeconomic impacts by comparing reduced healthcare costs with health gains and stratifying results by age, gender, and urban or rural residence for targeted insights. This approach addresses a methodological gap and provides evidence-based guidance to inform strategies for reducing Mongolia's burden of NCDs.

## Methodology and approach

### Study design overview

This study will employ a closed cohort Markov model for cost-effectiveness analysis from societal and healthcare perspectives. The model will simulate the Mongolian population, by age group, to capture variations in SSB consumption and health outcomes. Each age cohort will enter the model based on its baseline prevalence of NCDs, reflecting real-world conditions. The health states include 'healthy', 'NCD', and 'dead', with 'dead' serving as the absorbing state.

Fig 1 illustrates the relationships between the three health states, highlighting the transitions that occur within cohorts over time. Short-term symptoms, hospitalisation, or acute

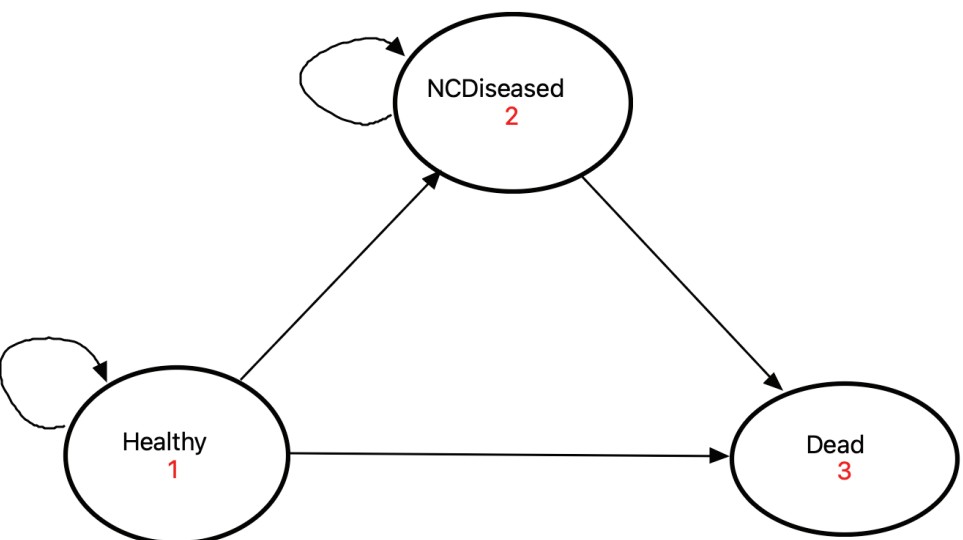

**Fig 1. Health state transition diagram**. This diagram visualises the relationships between the three health states.

conditions were not modelled as separate states, as they do not persist long enough to warrant separate state representation. Key transitions include movement from healthy to NCD and mortality across all states. Hospitalisation was not included as a separate state due to limited data on transition probabilities by hospitalisation status.

The key intervention will be a 'with taxation' on SSBs, and the comparator will be a 'no intervention' scenario, representing the status quo. Baseline characteristics, health costs, and transition probabilities will be derived from Mongolian surveys and international sources.

## Analytical approach

This study protocol outlines an ongoing project. Data collection is currently underway and is expected to be completed by March 2025. Data analysis will commence in April 2025, with preliminary results anticipated by July 2025. The study will integrate advanced analytical techniques to evaluate the impact of the SSB tax. A closed cohort Markov model developed in TreeAge Pro, will simulate disease progression and estimate the cost-effectiveness of the tax over a lifetime. The model will project changes in health outcomes, healthcare costs, and economic impacts. The Markov model includes three health states: Healthy (individuals without an NCD), NCD(individuals diagnosed with an obesity-related disease such as diabetes or cardiovascular disease), and Death (an absorbing state). Individuals transition between these states annually based on age-specific probabilities. Transitions from Healthy to NCD will be determined using incidence rates from epidemiological data, while NCD-to-Death transitions will incorporate disease-specific and all-cause mortality rates from [21]. Population groups are assumed to maintain their initial characteristics within the evaluation period, with no transitions between demographic categories. In addition, the model assumes that NCDs are irreversible, meaning individuals remain in the disease state once they transition into it [22]. Fig 2 depicts the Markov model framework created in TreeAge Pro, projecting health and economic outcomes over the study horizon. A half-cycle correction will be applied in TreeAge Pro by assigning 50% of the annual QALYs, costs, and other outcomes in the initial and final

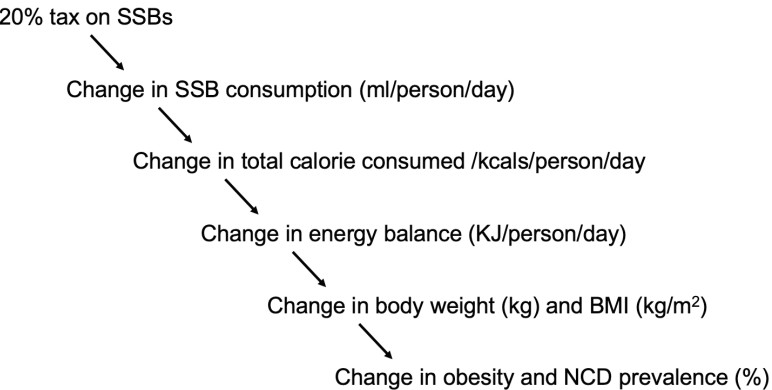

**Fig 2. Markov cohort model diagram.** This diagram represents the framework for simulating health states and transitions.

cycles to reflect mid-cycle transitions. We follow the Consolidated Health Economic Evaluation Reporting Standards checklist (CHEERS) and the PALISADE checklist, the guidance for best practices in applying machine learning in health economics and outcomes research, for transparent reporting [23,24]. Completed checklists are provided in S1 and S2 Tables.

The model will integrate data from multiple sources. The Mongolia STEPwise approach to noncommunicable disease risk factor surveillance (STEPS) survey 2019 [25], implemented by the Public Health Institute with the support of the World Health Organization, will serve as the primary health dataset. It provides detailed information on disease prevalence, BMI distribution, and healthcare costs, which are essential inputs for the analysis. To align with the latest demographic structure, we will adopt the 2023 Mongolian population dataset. All cost estimates from STEPS 2019 will be inflation-adjusted to 2023, with disease prevalence assumed unchanged due to a lack of updated data. Additional input sources include the Sixth National Nutrition Survey [26] for age-specific SSB consumption patterns, the Household Socioeconomic Survey [27] for socioeconomic factors and SSB pricing, and population data from the National Statistical Database [28].

Key parameters (see Table 1) include baseline characteristics, health-related and intervention costs, and transition probabilities, incorporating data from Mongolian surveys and international sources.

The effect of SSB taxation will be modelled by modifying transition probabilities from Healthy to NCD, reflecting reductions in SSB consumption and obesity-related disease risk. Pre-tax transition probabilities will be estimated based on age, and post-tax adjustments will be incorporated to evaluate potential changes in NCD incidence. The impact of taxation may vary across age groups, with younger populations potentially experiencing greater reductions in disease risk due to early-life behavioural changes [32,33]. The model will account for trends in obesity prevalence, assessing whether these rates remain stable or decline following the implementation of the tax. If obesity prevalence remains constant, the primary effect of the tax will be reflected in lower NCD incidence rather than changes in overall obesity distribution. However, if SSB consumption reductions lead to gradual weight loss, a corresponding decrease in obesity prevalence may further enhance the tax's long-term health benefits. Sensitivity analyses will test these assumptions and their influence on cost-effectiveness estimates. The tax is expected to reduce daily SSB intake, leading to caloric intake reduction, leading to changes in body weight and BMI over time. Using these BMI shifts, we will estimate

**Table 1. Description of key parameter inputs.**

| Data inputs | Description | Source |
|---|---|---|
| **Baseline characteristics** | | |
| Target population | Mongolian population 2023, aged over 15 years old | [27] |
| Baseline demographics | Age, gender, socioeconomic data | [25] |
| Mean daily SSB consumption | Estimated consumption per person | [26] |
| Consumption-weight relationship | 100-kJ/day change = 1-kg weight change | [29] |
| Body mass index | Mean and distribution of BMI | [25] |
| Disease epidemiology | Obesity-related disease prevalence | [25] |
| **Health costs from government** | | |
| Formal healthcare insurance costs | Annual per capita cost | [30] |
| **Health-related out-of-pocket costs** | | |
| Formal healthcare costs included | Fees for providers, medicines, and tests related to NCD treatment | [25] |
| Informal care costs | Lost work and transportation | [25,28] |
| **Intervention costs of SSB tax** | | |
| Government tax collection costs | 1% of SSB tax revenue | [31] |
| Industry compliance and reformulation costs | 1% of SSB tax revenue | [25] |

Note: SSB = Sugar-Sweetened Beverages; NCD = Non-communicable disease; BMI=Body Mass Index.

the potential impact on NCD risk reduction. Fig 3 illustrates the logical pathway for modelling the effects of an SSB tax, from tax implementation to health outcomes. A reduction in energy intake through lower SSB consumption is associated with small but cumulative weight changes over time, which in turn affects disease risk and long-term health outcomes. Post-tax adjustments will be explored using DML-estimated price elasticity to assess the tax's impact on SSB consumption and weight change.

Double Machine Learning (DML) will be employed to estimate the causal effects of the SSB tax on consumption and weight change while addressing potential confounding factors. Leveraging machine learning algorithms effectively reduces bias arising from unobserved heterogeneity and ensures robust estimation of treatment effects [20]. DML will be used to estimate price elasticity, accounting for confounding variables such as income, education, and pre-existing dietary behaviours [34]. We will adapt the analytical approach recommended by [35], carefully evaluating the methodological considerations of applying DML to our research design. The estimated changes in SSB consumption from DML will be incorporated into the Markov model by modifying transition probabilities from Healthy to NCD, reflecting the indirect impact of taxation on obesity-related diseases.

The DML approach will be implemented in Stata using the ddml command, which facilitates the estimation of causal effects by combining machine learning methods such as LASSO (Least Absolute Shrinkage and Selection Operator) and random forests with traditional econometric approaches. The ddml command applies LASSO for variable selection and Random Forest for non-linearities, then estimates the policy's causal effect by partially out confounders [36]. To enhance interpretability, we will apply SHAP (Shapley Additive Explanations) to the Random Forest model, summarising the impact of key predictors on price elasticity estimates and highlighting their contribution to predicted consumption changes, following the approach of [37]. The DML-estimated changes in SSB consumption will be integrated into the Markov model, modifying transition probabilities to reflect realistic consumption shifts and obesity-related health impacts.

The analysis will estimate the Incremental Cost-Effectiveness Ratio (ICER), expressed as the cost per QALY gained, DALYs averted, tax revenue generated, and reductions in disease

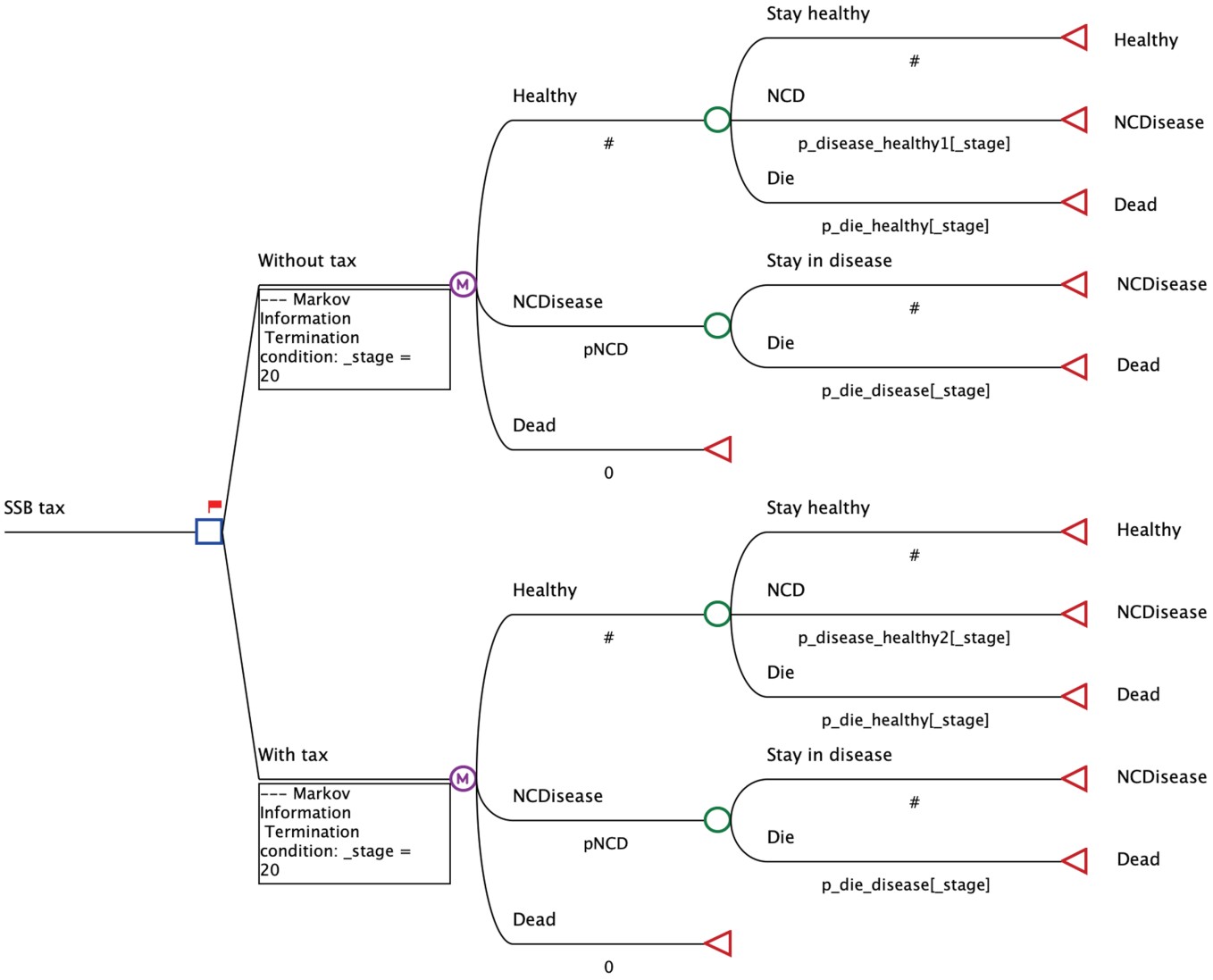

**Fig 3. Pathway of estimated effect of SSB tax on BMI through consumption changes.**

burden.QALY and DALY estimates for NCD cases are sourced from [38], with a disability weight of 0.26 for DALYs and a utility weight of 0.74 for QALYs. These values inform the Markov model, linking SSB-induced BMI reductions to obesity-related disease risks. Transition probabilities will be adjusted using DML-based estimates, and the model tracks health outcomes over the lifetime of the cohort.

The economic evaluation will include direct healthcare costs (e.g., treatment for cardiovascular disease and diabetes) and indirect costs (e.g., lost productivity).

## Population

This study models the 2023 Mongolian population aged over 15 from the National Statistical Office [27]. The 2019 STEPS survey, our primary dataset for health status and disease prevalence, only includes individuals aged 15 and older, making this the most appropriate

population for analysis. Since NCDs such as obesity, diabetes, and cardiovascular disease primarily affect adults and younger age groups are not covered in STEPS, extending the model to children would require additional assumptions beyond the available data. Mortality rates will be sourced from [21], with one-year cycles modelled in TreeAge Pro 2022.

## Time horizon

The time horizons will be set at 20 years and lifetime to capture the medium and long-term impacts of the SSB tax in Mongolia.

## Intervention

The primary intervention is a 20% excise tax on SSBs, following WHO recommendations, applied to all non-alcoholic beverages with added sugars, excluding pure fruit juices and milk-based products. This intervention will be modelled as a one-time implementation at the beginning of the simulation period and will remain effective throughout the 20-year analysis. The comparator will be a "no intervention" scenario, representing the status quo, accounting for potential shifts in consumption and industry responses (e.g., reformulation).

## Outcomes

The primary outcome of this study is the Incremental Cost-Effectiveness Ratio (ICER), expressed as the cost per QALY gained. The ICER will quantify the cost-effectiveness of implementing a 20% SSB tax in Mongolia by comparing the incremental costs and health benefits of the tax relative to a scenario without it. The ICER is calculated as:

$$\text{ICER} = \frac{\Delta\text{Cost}}{\Delta\text{QALY}} \tag{1}$$

where

- $\Delta$Cost = Difference in total costs between the intervention (SSB tax) and the baseline (no tax).
- $\Delta$QALY = Difference in QALYs gained between the intervention and baseline scenarios.

Secondary outcomes include a range of health, economic, and social indicators. Health outcomes encompass DALYs averted, QALYs gained, life years gained, changes in Body Mass Index (BMI) distribution, and the incidence and prevalence of obesity-related diseases. Economic outcomes include variations in healthcare costs, productivity, and revenue generated from taxes. Changes in SSB consumption patterns and potential for increased adoption of healthier alternatives will also be assessed. Furthermore, the analysis will consider industry impacts, including changes in SSB sales volumes, revenue, and the rate of product reformulation.

## Assessment of costs

The costs associated with the SSB tax policy will include both direct and indirect costs. Direct costs will account for administrative expenses such as tax collection, compliance monitoring, and public awareness campaigns. These will be estimated as a percentage of the total SSB tax revenue, based on methodologies employed in similar studies [39,40]. However, the official Mongolian tax policy document states that there will be no additional costs for tax collection

and implementation, as existing structures will be utilised [41]. We adopt a tax collection cost of approximately 1% of revenue, along with industry compliance costs based on prior literature [9]. Sensitivity analyses will be conducted to test the impact of varying implementation costs on cost-effectiveness estimates. Healthcare costs are expected to decrease as reduced SSB consumption leads to a lower incidence of NCDs. This reduction in healthcare expenditures will be factored into the overall net cost calculation. Indirect costs will include industry compliance expenses, such as those incurred by beverage manufacturers for reformulating products, updating labelling, and modifying advertising strategies to meet new regulations. Based on previous studies, compliance and reformulation costs will be estimated at 1% of tax revenue [9]. Additionally, the study will also look at productivity losses in the beverage industry. It will consider the opportunity costs of using resources for this intervention instead of other public health measures [42].

## Assessment of benefits

The benefit assessment for the tax will evaluate its health, economic, and social impacts. The tax is expected to reduce the incidence and prevalence of obesity, type 2 diabetes, cardiovascular diseases, and other NCDs linked to high SSB consumption. These impacts are measured in terms of DALYs averted, QALYs gained, and life years gained, reflecting improvements in population health, life expectancy, and overall well-being [1,2]. Lower disease rates are anticipated to result in healthcare cost savings by reducing the need for medical services, such as hospitalisations, medications, and clinical consultations [9]. Additionally, healthier populations are likely to experience increased productivity due to fewer workdays lost and lower disability rates, with productivity gains estimated based on Mongolian labour productivity metrics [7]. Meanwhile, by making SSBs less affordable, the tax may encourage healthier beverage choices, such as water or low-sugar alternatives, with consumption shifts estimated through projected reductions in SSB intake and potential substitution effects [8]. Revenue generated from the tax could be reinvested into public health initiatives, creating a cycle of health improvements that further amplify the policy's benefits [9,10].

## Discount rate

Following the norm in the literature, a 5% discount rate will be applied to costs and health outcomes from year 2 onward, in line with recommendations for economic evaluations in low- and lower-middle-income countries [43]. This rate reflects Mongolia's lower-middle-income status in 2023, the baseline year for our population data and economic analysis. All costs and outcomes will be adjusted to 2023 values. All monetary values will be adjusted to 2023 Mongolian Tugrik and converted to USD using the 2023 exchange rate. To test robustness, sensitivity analyses will be conducted using alternative rates of 3% and 7%.

## Sensitivity analysis

The robustness of the cost-effectiveness of the proposed tax will be assessed using deterministic and probabilistic sensitivity analysis, to account for the uncertainty related to parameters and observed variables [44]. In the deterministic analysis, key parameters—such as elasticity of demand, intervention decay rate, healthcare costs, SSB consumption, and industry reformulation will be varied individually to determine their influence on outcomes. Additionally, alternative scenarios will be explored, considering different time horizons, discount rates, pass-through effects, and tax structures to assess their influence on the results. Probabilistic

sensitivity analysis will use Monte Carlo simulations to produce confidence intervals and cost-effectiveness acceptability curves (CEACs).

## Model validation

The model will undergo several validation steps to ensure accuracy and reliability. Expert consultation will confirm that the model aligns with the Mongolian context. Internal validation will check parameter consistency, particularly for key variables such as price elasticity of demand. Cross-validation will compare model outputs with findings from similar studies to ensure alignment with established patterns. Calibration against Mongolia-specific baseline data will validate the model's initial conditions. All assumptions and limitations will be clearly documented.

## Equity considerations

The study will examine how the SSB tax affects health outcomes across different demographic and socioeconomic groups in Mongolia. Distributional Cost-Effectiveness Analysis (DCEA) will be used to assess whether the tax reduces health inequalities, focusing on groups with higher baseline health risks, such as lower-income or rural populations. This approach will help determine how much the tax can promote more equitable health benefits.

## Ethics

This study exclusively utilised publicly available secondary data and involved no direct interaction with human subjects. According to the updated National Statement on Ethical Conduct in Human Research (2023), a risk assessment was conducted to confirm that the study is exempt from further ethical review.

## Limitation

This study has several limitations. It relies on secondary data sources, including the STEPS survey, National Nutrition Survey, and Household Socioeconomic Survey, which may not fully capture all factors influencing SSB consumption and health outcomes in Mongolia. The closed cohort Markov model assumes no demographic changes during the evaluation period, potentially affecting long-term projections. While DML improves causal inference, residual confounding from complex non-linear relationships remains possible. Additionally, broader social and industry-level effects, such as employment changes in the beverage sector, are not included in the analysis. Sensitivity analyses will address these limitations by testing result robustness under different scenarios. In addition, cross-price elasticity effects on beverage substitution will not be considered due to data limitations.

## Conclusion

This study combines economic modelling and simulation techniques to evaluate the health impacts, cost-effectiveness, and equity implications of the proposed SSB tax in Mongolia. By employing DML alongside a Markov cohort model, the research offers a comprehensive assessment of the tax's impact on consumption, obesity-related diseases, and healthcare costs. This approach addresses key gaps in previous studies and applies innovative techniques to a low- and middle-income country context, where evidence on SSB taxation remains scarce. The findings will provide Mongolian policymakers with valuable, evidence-based insights to guide their efforts in reducing the burden of NCDs and promoting fairer health outcomes for all.

## Supporting information

**S1 Table. CHEERS checklist.**
(DOCX)

**S2 Table. PALISADE checklist.**
(DOCX)

## Author contributions

**Conceptualization:** Nyamdavaa Byambadorj, Undram Mandakh, Kompal Sinha.

**Data curation:** Nyamdavaa Byambadorj.

**Formal analysis:** Nyamdavaa Byambadorj.

**Investigation:** Nyamdavaa Byambadorj, Undram Mandakh.

**Methodology:** Nyamdavaa Byambadorj, Rohan Best.

**Software:** Nyamdavaa Byambadorj.

**Supervision:** Rohan Best, Undram Mandakh, Kompal Sinha.

**Validation:** Kompal Sinha.

**Visualization:** Nyamdavaa Byambadorj.

**Writing – original draft:** Nyamdavaa Byambadorj.

**Writing – review & editing:** Rohan Best, Undram Mandakh, Kompal Sinha.

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
