## [Decision Letter · Decision Letter 0]

5 Mar 2025

PONE-D-25-01490Protocol for evaluating the Cost-Effectiveness of Mongolia's Sugar-Sweetened Beverages Tax using Double Machine LearningPLOS ONE

Dear Dr. Byambadorj,

Thank you for submitting your manuscript to PLOS ONE. After careful consideration, we feel that it has merit but does not fully meet PLOS ONE’s publication criteria as it currently stands. Therefore, we invite you to submit a revised version of the manuscript that addresses the points raised during the review process.

We look forward to receiving your revised manuscript.

Kind regards,

Francis Xavier Kasujja

Academic Editor

PLOS ONE

Reviewers' comments:

Reviewer's Responses to Questions

**Comments to the Author**

1. Does the manuscript provide a valid rationale for the proposed study, with clearly identified and justified research questions?

Reviewer #1: Yes

Reviewer #2: Yes

2. Is the protocol technically sound and planned in a manner that will lead to a meaningful outcome and allow testing the stated hypotheses?

Reviewer #1: Yes

Reviewer #2: Partly

3. Is the methodology feasible and described in sufficient detail to allow the work to be replicable?

Reviewer #1: Yes

Reviewer #2: No

4. Have the authors described where all data underlying the findings will be made available when the study is complete?

Reviewer #1: Yes

Reviewer #2: No

5. Is the manuscript presented in an intelligible fashion and written in standard English?

Reviewer #1: Yes

Reviewer #2: Yes

6. Review Comments to the Author

You may also provide optional suggestions and comments to authors that they might find helpful in planning their study.

Reviewer #1: I have reviewed the manuscript and am pleased with the detailed explanation and its quality. I hope the study went well and look forward to reading the final paper.

Reviewer #2: Protocol for evaluating the Cost-Effectiveness of Mongolia’s Sugar-Sweetened Beverages Tax using Double Machine Learning

The paper aims to estimate the cost-effectiveness of the SSB tax on health outcomes and savings in Mongolia. The study employs a Markov cohort model combined with Double Machine Learning. The proposed analytical approach is methodologically sound and innovative, integrating economic modeling with causal inference techniques. However, the protocol lacks a sufficiently detailed description of the methods, which is crucial for assessing the robustness and feasibility of the study. By addressing these methodological concerns and enhancing transparency in model specification, data sources, this research could provide stronger evidence to inform public health policy in Mongolia and beyond.

Abstract

1. Authors say “Given the limited research on the causal relationship between SSB consumption and health outcomes is scarce, we conduct a Cost-Effectiveness Analysis (CEA) using a Markov cohort model and Double Machine Learning (DML) to control for confounding factors and refine causal estimates”.

Nowadays a lot of evidence exist about the causal effect of SSB on obesity-related diseases.

Data inputs and model specification

1. To strengthen the study, it would be beneficial to include a figure illustrating the overall modeling process. For example, the change in kcal will be modeled using own price elasticities. For change in obesity,… so the reader is engaged and interested in the methodology.

2. Model Specification and Justification: The protocol should provide a more comprehensive description of the Markov cohort model. Some information should be detailed as follows:

o “Disease” in the Markov model is obesity or obesity-related diseases?

o Describe the state transition probabilities, and assumptions regarding disease progression, mortality and remission. It would be good to include these parameters in table 1.

o Mortality rates come from GBD data source, that will conclude in February 2024, but this date already passed.

3. Is there information about hospitalization of people with obesity?

4. Trends in obesity prevalence will be considered, or it will be constant over time?

5. It should clarify how DML is integrated within the modeling framework and how DML will control for confounding?

6. It would be worthwhile to write a paragraph describing Stata command: the ‘ddml’ command.”

7. I went to ref 23 and I could not find information on disease costs. Please add this information in an appendix.

8. Proportion of SSB intake by population: 44.5% I don´t understand this input. Nearly 50% of total energy intake comes from SSBs? Sounds too much.

9. Target population: 0-100 years. In other sections it says 2-100.

10. The cost of implementing SSB tax in Mongolia is from an US reference as a % of the SSB tax revenue. While it was done previously, a discussion about this assumption should be presented.

11. An analysis through performance indices would be interesting. With it, you could clarify how a model such as the Random Forest can capture model nonlinearities compared to LASSO.

12. It would be worth considering techniques such as SHAP (Shapley Additive Explanations), as they facilitate the interpretation of the model results, summarizing the impact of each characteristic on the prediction and how each variable contributes to the predictions.

Assessment of cost

I could not assess how healthcare expenditures will be estimated.

Assessment of benefit

I could not assess how DALYs and QALYs will be included in the model.

Minor changes

It would be worthwhile adding the definition of the acronym LASSO (Least Absolute Shrinkage and Selection Operator) before mentioning the acronym.

7. PLOS authors have the option to publish the peer review history of their article (what does this mean?). If published, this will include your full peer review and any attached files.

Reviewer #1: **Yes: **Abdillah Ahsan

Reviewer #2: **Yes: **Ana Basto-Abreu

---

## [Author Response · Author response to Decision Letter 1]

7 Apr 2025

Thank you for the constructive feedback. We have addressed all comments in the attached Response to Reviewers document and updated the manuscript accordingly (with and without track changes).

---

## [Editor Report · Decision Letter 1]

24 Apr 2025

Protocol for evaluating the Cost-Effectiveness of Mongolia's Sugar-Sweetened Beverages Tax using Double Machine Learning

PONE-D-25-01490R1

Dear Dr. Byambadorj,

We’re pleased to inform you that your manuscript has been judged scientifically suitable for publication and will be formally accepted for publication once it meets all outstanding technical requirements.

Kind regards,

Francis Xavier Kasujja

Academic Editor

PLOS ONE

---

## [Editor Report · Acceptance letter]

PONE-D-25-01490R1

PLOS ONE

Dear Dr. Byambadorj,

I'm pleased to inform you that your manuscript has been deemed suitable for publication in PLOS ONE. Congratulations! Your manuscript is now being handed over to our production team.

Kind regards,

on behalf of

Dr. Francis Xavier Kasujja

Academic Editor

PLOS ONE